# A dual fluorescent *Plasmodium cynomolgi* reporter line reveals in vitro malaria hypnozoite reactivation

Annemarie M. Voorberg-van der Wel [1], Anne-Marie Zeeman[1], Ivonne G. Nieuwenhuis[1], Nicole M. van der Werff[1], Els J. Klooster[1], Onny Klop[1], Lars C. Vermaat[1], Devendra Kumar Gupta[2], Laurent Dembele[2,3], Thierry T. Diagana[2] & Clemens H.M. Kocken[1]*

*Plasmodium vivax* malaria is characterized by repeated episodes of blood stage infection (relapses) resulting from activation of dormant stages in the liver, so-called hypnozoites. Transition of hypnozoites into developing schizonts has never been observed. A barrier for studying this has been the lack of a system in which to monitor growth of liver stages. Here, exploiting the unique strengths of the simian hypnozoite model *P. cynomolgi*, we have developed green-fluorescent (GFP) hypnozoites that turn on red-fluorescent (mCherry) upon activation. The transgenic parasites show full liver stage development, including merozoite release and red blood cell infection. We demonstrate that individual hypnozoites actually can activate and resume development after prolonged culture, providing the last missing evidence of the hypnozoite theory of relapse. The few events identified indicate that hypnozoite activation in vitro is infrequent. This system will further our understanding of the mechanisms of hypnozoite activation and may facilitate drug discovery approaches.

[1] Department of Parasitology, Biomedical Primate Research Centre, 2288 GJ Rijswijk, The Netherlands. [2] Novartis Institute for Tropical Diseases, Emeryville, CA, USA. [3]Present address: Faculty of Pharmacy, Université des Sciences, des Techniques et des Technologies de Bamako (USTTB), MRTC – DEAP, Bamako, Mali. *email: kocken@bprc.nl

 **1**

The majority of *Plasmodium vivax* infections is likely due to relapses[1]. Not to be mixed up with recurrences, which may have other origins[2], relapse infections have been posited to derive from hypnozoites[3]. After the identification of hypnozoites in livers infected with *Plasmodium cynomolgi*[4,5] and *P. vivax*[6], the hypnozoite theory of relapse has been questioned[7–9]. One of the arguments was that the actual transformation of the hypnozoite into a schizont had never been observed[7,10]. The currently available in vitro models using nongenetically modified parasites all have in common that hypnozoites and developing forms are identified on the basis of morphological features in fixed tissue cells. As a result, it is impossible to monitor the growth of individual parasites over time, precluding direct observations of the actual transition of a hypnozoite into a developing parasite.

To overcome this hurdle, we here describe the construction of a *P. cynomolgi* M strain dual fluorescent reporter line that can distinguish hypnozoites (green-fluorescent protein, GFP positive and mCherry negative) from developing forms (both GFP and mCherry positive). Using this transgenic parasite line, we show direct proof for reactivation of dormant hypnozoites in real time. This demonstrates, almost 40 years after their discovery, that hypnozoites have the capacity to awaken to resume liver stage development, which in vivo ultimately provokes malaria relapses.

## Results

**Transgenic *P. cynomolgi* recapitulates the full life cycle**. To express the fluorescent reporters GFP and mCherry in liver stage parasites, a *P. cynomolgi* reporter line was constructed using a centromere plasmid that is maintained throughout the life cycle[11,12] and a human dihydrofolate reductase (Hdhfr) selectable marker[13] controlled by the constitutive *P. cynomolgi* hsp70 promoter. Through the inclusion of a T2A self-cleaving peptide[14] not only hdhfr but also gfp was driven by the hsp70 promoter. In addition, to highlight stages that have initiated schizogony, the construct contained mCherry controlled by promoter and 3′UTR of the recently described schizont-specific marker for early liver stage growth, *P. cynomolgi* liver-specific protein 2 (lisp2)[15] (Fig. 1a; see Methods section for details). Following transfection[16], pyrimethamine-resistant parasites emerged at a growth rate comparable to wild-type parasites (Supplementary Fig. 1a). Ex vivo blood stage parasites showed strong GFP expression, while mCherry expression, was not observed (Supplementary Fig. 1b), recapitulating the expression of the LISP2 protein[15]. When ex vivo blood stage schizonts and merozoites were imaged for prolonged times, merozoite invasion of rhesus red blood cell (RBC) was observed (Supplementary movie 1). After feeding mosquitoes using a glass-feeder system, the transmission characteristics were determined. A side by side comparison of the transgenic line to wild-type parasites showed similar numbers of oocysts and sporozoites (Supplementary Fig. 1c). Gametocytes expressing GFP were readily observed in the blood used for mosquito feeding and upon incubation at room temperature (RT) for 10–15 min the process of exflagellation could be viewed in real time (Supplementary movie 2). Following mosquito transmission, primary rhesus hepatocytes were infected with sporozoites. A side by side comparison of wild type and transgenic liver stage parasites at day 6 of development by immunofluorescence assay (IFA)[17] showed that numbers of exoerythrocytic forms (EEFs) and hypnozoite ratio of the transgenic line were similar to wild-type parasites (Fig. 1b). Beyond day 6, parasite development appeared to progress normally and schizonts had fully matured around day 10 post sporozoite invasion, at which time merosomes could be observed (Fig. 1c and Supplementary movie 3). To determine whether the full life cycle could be recapitulated, rhesus RBCs were added to wells that contained merosomes and incubated overnight. The next day, blood stage parasites were observed in these wells by Giemsa staining

(Fig. 1c). Taken together, these data indicate that the transgene expression of fluorescent proteins did not affect parasite development in any stage of the life cycle, not only providing the opportunity to visualize essential parasitic processes throughout the parasite life cycle, but also to study hypnozoite activation using this parasite line.

To determine the percentage of liver stage parasites that had lost the centromere plasmid as a result of the many nuclear divisions that take place at the various stages of the life cycle in the absence of pyrimethamine selection[18], live images of hepatocyte cultures were acquired followed by the fixation and IFA of the same wells. Indeed, GFP signals obtained were specific for liver stage parasites and both hypnozoites and schizonts expressed GFP (Fig. 1d). GFP signals were observed in 73% of the parasites as determined by IFA (counting > 200 parasites), indicating that 27% of the liver stage parasites had lost the centromere plasmid. This is similar to what has previously been reported for centromere plasmids[11,12].

**mCherry is specifically expressed in liver schizonts**. To observe the timing of GFP and mCherry expression, live fluorescence microscopy at a high magnification (63×; 1.40 NA) was performed at different time points of liver stage development. GFP was expressed throughout the liver stage development in both small and large forms (Fig. 2). At day 1 post sporozoite infection, round liver stage forms were observed. In addition, some sporozoites had not yet fully transitioned into rounded forms and the two distal ends of the sporozoites could still be observed. Similar to what has been described before for LISP2 expression[15], lisp2-driven mCherry expression was absent in the first two days post sporozoite infection. In a minor parasite fraction at day 3 post sporozoite invasion mCherry expression first emerged, often coinciding with a slightly more pronounced growth in size as compared to small forms (Fig. 2). From then onward, an mCherry-positive population that grew in size and became multinucleated could be differentiated from an mCherry-negative population that remained single nucleated and small in size. During schizont maturation, mCherry expression levels rapidly increased with peak expression around days 8–10 (Fig. 2). mCherry fluorescence remained visible in merosomes. To determine whether mCherry expression coincided with nuclear division, at different time points post sporozoite infection, liver stage cultures were fixed and 4′,6-diamidino-2-phenylindole, dilactate (DAPI, dilactate) was added for nuclear staining. GFP and mCherry signals could still be observed after fixation allowing counting of the number of GFP- and mCherry-positive parasites at each time point in relation to the number of parasite nuclei. The onset of mCherry expression corresponded with the initiation of nuclear division, as all GFP-positive parasites with >1 nucleus were mCherry positive (Table 1 and Supplementary Fig. 2). These data indicate that the regulatory sequences used in the centromeric plasmid contained sufficient information for the correct timing of lisp2-driven mCherry expression[19].

We also noted that cultures at days 2 and 3 contained substantially more parasites than cultures at later time points (Table 1). The loss of parasites during the initial stages of parasite development has been described before[15,20] and may be a result of cytosolic immune responses, such as selective autophagy by the host cell[20] or could reflect a parasite developmental failure.

**Long-term live imaging of *P. cynomolgi* liver stage parasites**. Following the characterization of the parasites at high magnification, our next aim was to visualize the development of individual parasites over time. We initiated long-term liver stage cultures with the transgenic *P. cynomolgi* line in 96-well plates

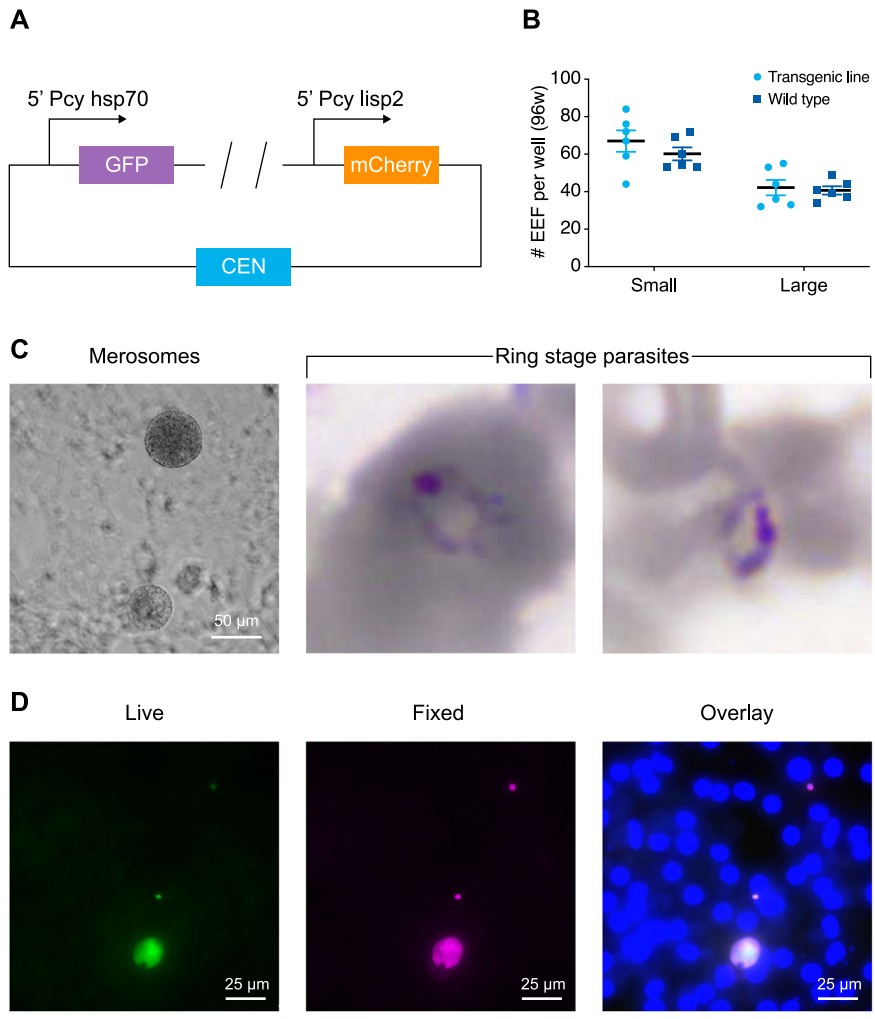

**Fig. 1 The *P. cynomolgi* dual reporter line fully recapitulates the complete life cycle. a** Schematic of the most important building blocks of the construct used for transfection; for details of the complete construct, see methods and the Supplementary Data file 1. The plasmid contains two fluorescent reporters, GFP driven by the *hsp70* promoter and mCherry controlled by the *lisp2* promoter. A centromere is included to maintain the construct throughout the life cycle. **b** Dot plot of the number of small and large exoerythrocytic forms (EEFs) per well in a 96-well plate with s.e.m. from *n* = 6 wells. Results derive from a side by side comparison of transgenic and wild-type parasites. **c** Left, *P. cynomolgi* merosomes (transgenic line) in culture at day 10 post sporozoite invasion. Scale bar, 50 µm. Right, Giemsa staining of red blood cells collected one day post overlay on a liver stage culture containing merosomes shows two ring stage blood forms. **d** GFP expression (live) in two small forms and one large form at day 6 post inoculation of a primary hepatocyte culture with transgenic *P. cynomolgi* sporozoites; the same image after fixation and IFA with anti-HSP70 antibodies, and an overlay with DAPI. Scale bars, 25 µm.

using a Matrigel cover[21] and performed daily or every other day imaging at a lower magnification using an Operetta high content imager (PerkinElmer). When the development of parasites within the same well was monitored between days 8–15 post sporozoite infection, well overviews made from stitched images revealed that the majority of the schizonts burst in vitro between days 10 and 13 (Fig. 3a and Supplementary movie 4). This closely mimics the maturation of *P. cynomolgi* EEFs in vivo, which are mature between days 9 and 12[22]. The development of individual parasites could be monitored in more detail in the nonstitched pictures, revealing GFP-positive parasites that remain small, and parasites (schizonts) that in addition become mCherry positive and progressively grow in size and eventually rupture after day 10 (Fig. 3b). Taken together, this shows that the dual fluorescent parasite line enables tracking of individual parasites in real time.

To obtain more quantitative data, the GFP and mCherry intensities, and the fluorescent surface area as a proxy for the parasite size of individual hypnozoites and schizonts present in the same ten fields of a well at different time points were measured using the Operetta. This showed that in schizonts mCherry levels increase over time, coinciding with parasite growth (Fig. 3c). A clear distinction is visible between a population with background levels for mCherry (hypnozoites) and an mCherry-positive population (schizonts). GFP expression also increases over time, but this does not allow a clear delineation of the transition between hypnozoites and schizonts, likely reflecting the constitutive nature of the *hsp70* promoter (Fig. 3c). The size and mCherry measurements indicate that at day 15 post sporozoite inoculation a number of schizonts are present (Fig. 3c). To further quantitate this, for four wells derived from three independent infections, the total number of hypnozoites and schizonts present at days 10 and 15 was determined (Supplementary Table 1). This revealed important issues regarding EEF kinetics. The total number of EEFs reduces over time with 50% on day 15 compared to day 10. Both the number of schizonts and the number of hypnozoites are reduced (29% and 66% present at day 15, respectively, as compared to day 10). This reduction is caused by culture technical reasons (parasite death and hepatocyte detachment), by mature schizont rupture and, possibly, by hypnozoite reactivation. The relatively

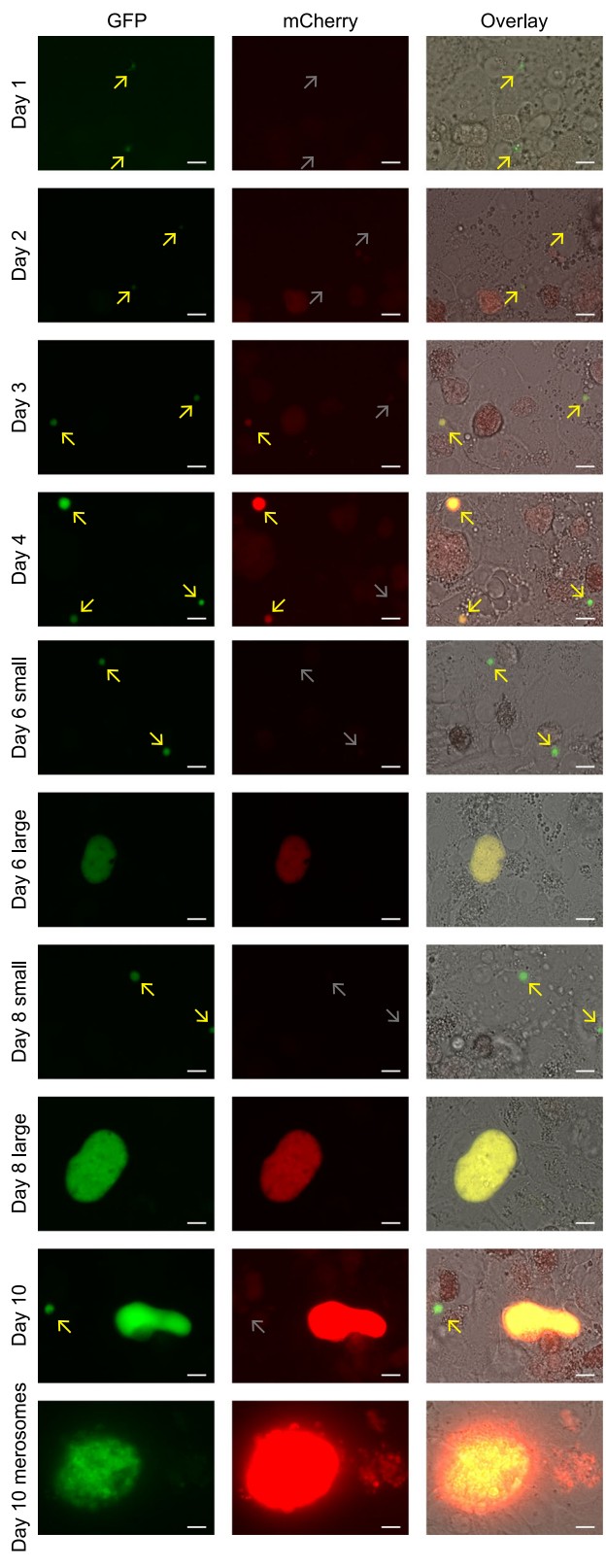

**Fig. 2** ***Lisp2*-driven mCherry differentiates *P. cynomolgi* hypnozoites from developing forms.** GFP and mCherry expression at different time points of transgenic *P. cynomolgi* liver stage development. The panels at the righthand side show an overlay with brightfield images. Scale bars, 10 μm.

**Table 1 Quantitation of schizont specific expression of mCherry.**

| Day p.i. | # Parasites (1n) | | # Parasites (>1n) | |
|---|---|---|---|---|
| | GFP +ve only | GFP and mCherry +ve | GFP +ve only | GFP and mCherry +ve |
| 2 | 170 | 0 | 0 | 0 |
| 3 | 143 | 1 | 0 | 0 |
| 4 | 71 | 9 | 0 | 24 |
| 6 | 58 | 1 | 0 | 37 |
| 10 | 36 | 3 | 0 | 19 |

Numbers of parasites in one well (96-well plate) expressing GFP only, or expressing both GFP and mCherry. Parasites were scored after fixation at different time points p.i. (post infection). Parasites were categorized as 1n (containing 1 nucleus) or >1n (multiple nuclei) as determined by the DAPI signal

high levels of schizonts we observed on day 15 have also been described by others using wild type *P. cynomolgi* EEF cultures[15], and 'late' schizonts have also been seen in liver biopsies from *P. cynomolgi*-infected monkeys[23]. The complexity of the culture dynamics precludes pinpointing where these late schizonts derive from in standard cultures. Our transgenic parasite, however, allowed us to trace back in time each late schizont, to determine whether they had originated from hypnozoite reactivation.

**Live visualization of *P. cynomolgi* hypnozoite activation.** In vivo, *P. cynomolgi* M is a fast relapsing parasite. While low-dose inoculations (2000 sporozoites) followed by subcurative treatment of blood stages gave rise to somewhat delayed relapses[24,25], high-dose sporozoite inoculations ($10^5$–$10^6$) resulted in early relapses around days 18–33 post infection as assessed by thin film analysis[26,27]. It is thus expected that activation of hypnozoites in this parasite may start relatively soon after the completion of liver stage development (merosome release) of the primary schizonts. To identify parasites that had activated, images were acquired daily or every other day. We then looked for the presence of schizonts in images taken at days 15–17 and compared this to images taken from the same field at day 10 post sporozoite infection. Considering that the bulk of the primary schizonts have reached full maturity at day 10, we chose this time point as baseline. Only parasites that had remained small and GFP positive, but mCherry negative at day 10 were posited to be hypnozoites. Parasites that were hypnozoites at day 10 and schizonts at days 15–17 were considered to originate from activated hypnozoites. This was further substantiated by monitoring parasite development between these time points. Following this methodology, we found that the overwhelming majority of schizonts found at day 15 had already been present at day 10 as well, indicating that in culture, schizonts can remain lingering on for prolonged periods of time. However, in a few instances we could indeed identify hypnozoites that initiate mCherry expression after day 10 and develop into schizonts. Moreover, we could visualize the full maturation of such a secondary schizont as highlighted by the budding of merosomes (Fig. 4a). When we overlaid the well overviews from different time points, generating a movie of parasite development, it was clear that the hypnozoite activation event took place after primary schizont development in the same well (Supplementary movie 4).

**Early in vitro hypnozoite activation occurs infrequent.** One of the hallmarks of hypnozoites is that they are resistant to days 5–8 treatment with the PI4K inhibitor KDU691[15,27]. To further prove that the events we observed are true hypnozoite activation events, and not retarded schizonts, we pretreated cultures on days 5–8

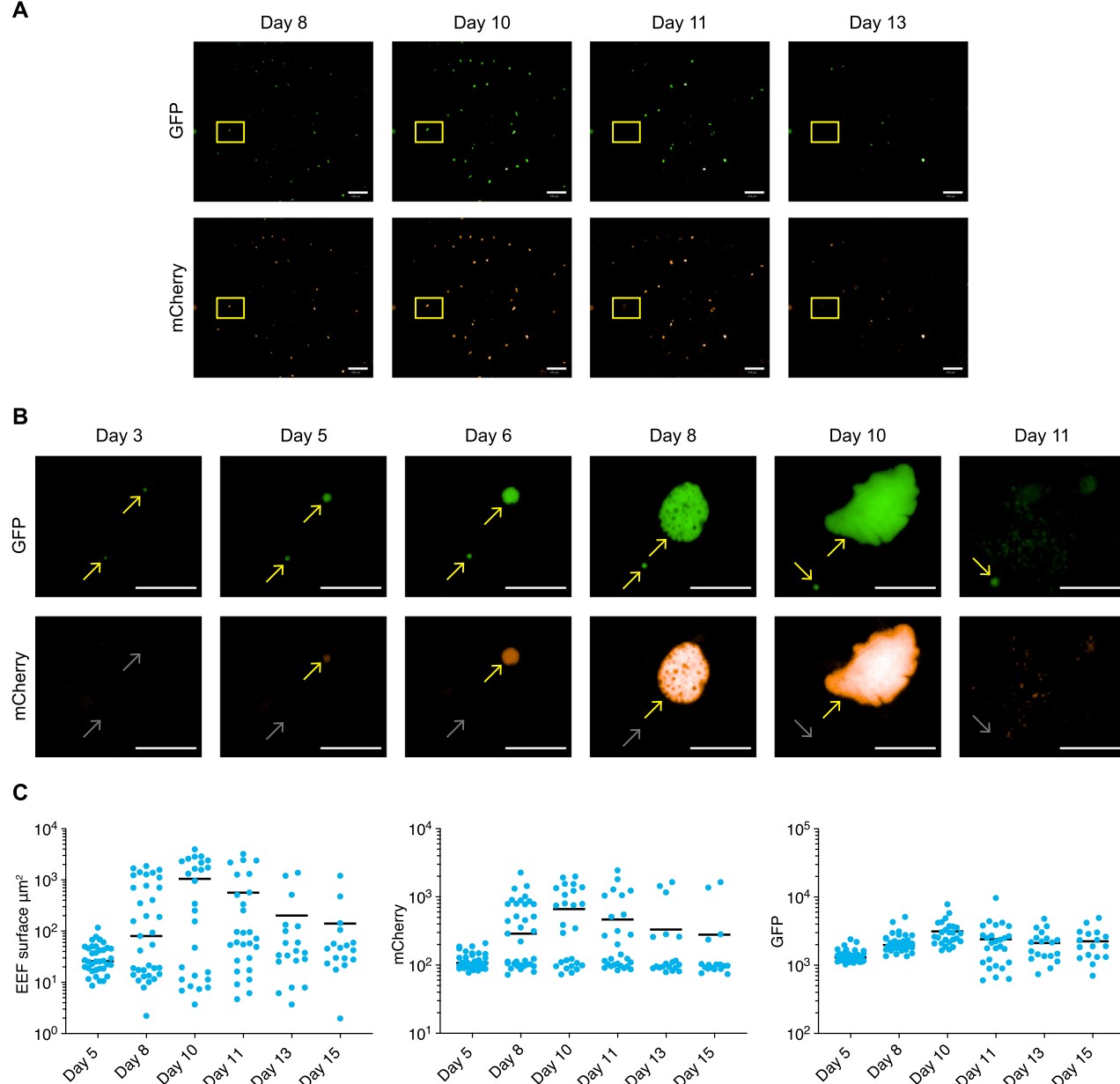

**Fig. 3 The dual fluorescent *P. cynomolgi* line reveals the real-time development of individual parasites. a** Images of parasites visualized live by fluorescence microscopy using an Operetta High Content imager. Overviews are shown from stitched images of the same well (96-well plate) at different time points, showing GFP-expressing parasites (upper panel) and mCherry-expressing parasites (lower panel). Parasites from the yellow inset are depicted in **b**. Scale bar, 500 μm. **b** Images of a small and a developing form expressing GFP (upper panel) and mCherry (lower panel) at different time points after sporozoite invasion. Yellow arrows mark expression, and grey arrows mark absence of fluorescence. At day 11, individual merozoites can be observed. Scale bar, 50 μm. **c** Operetta measurements of fluorescent EEF surface areas (left panel), mCherry intensity (middle panel), and GFP intensity (right panel) in individual parasites (blue dots) followed at different days p.i. (post infection) in ten fields of a single well. The bar represents the median values.

with KDU691 to eliminate schizonts from the cultures prior to expected hypnozoite reactivation. The treatment resulted in an almost complete removal of schizonts (2 schizonts left at day 9, compared to 21 schizonts in a nontreated well; Supplementary Fig. 3). When comparing day 17 images with day 11, the well overviews at day 17 revealed a schizont that was not present as schizont at day 11. Tracing this parasite back in time revealed the presence of a small form that started mCherry expression at day 13 post infection (Fig. 4b, white arrow). From that time onward, the parasite continues to grow, indicative of hypnozoite activation. We conclude from this that the fluorescent parasite line for the first time since hypnozoite discovery over three decades ago demonstrates the transition of hypnozoites into a schizont, providing strong evidence for the hypnozoite theory of relapse.

Next, in five independent infections we determined the number of spontaneously activated hypnozoites up to day 22, depending on length of cultivation. After scanning 21 untreated wells for reactivation events, corresponding to ±1175 day-15 hypnozoites, we observed only three events of activation (deriving from two independent infections). This indicates that spontaneous hypnozoite activation in vitro is an infrequent event during the first 3 weeks of culture.

To visualize the development of the observed events, mCherry intensities and EEF surfaces of the activated hypnozoites were

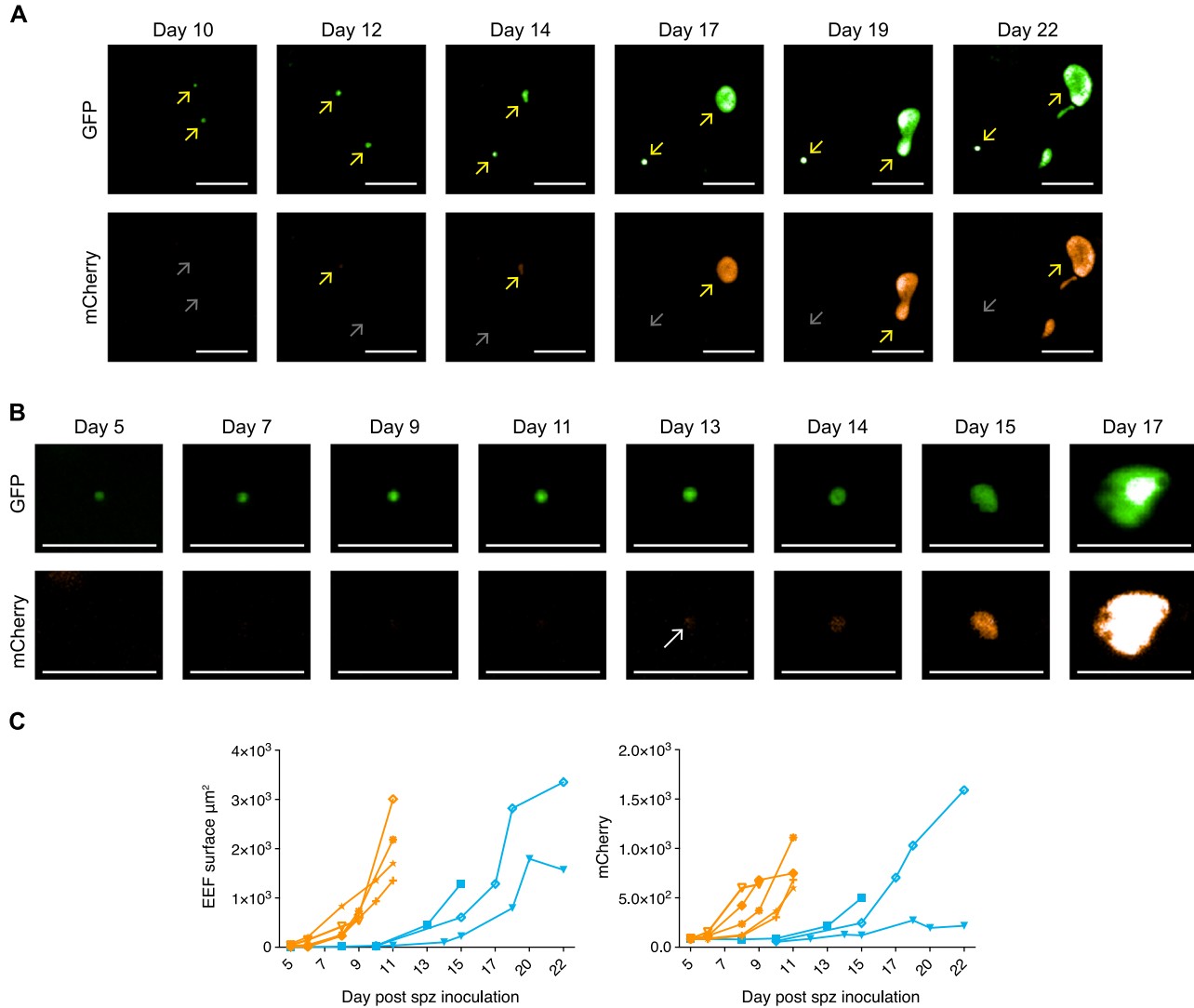

**Fig. 4 Monitoring individual parasites over time by fluorescence microscopy reveals hypnozoite reactivation. a** Hypnozoite reactivation resulting in a fully mature liver stage schizont releasing merosomes. Operetta images show GFP expression (upper panel) and mCherry expression (lower panel) of a hypnozoite (mCherry negative), and an activating hypnozoite at different time points after sporozoite invasion. Yellow arrows mark expression, and grey arrows mark absence of fluorescence. Scale bar, 50 μm. **b** Images showing GFP expression (upper panel) and mCherry expression (lower panel) in an activating hypnozoite at different time points after sporozoite invasion. A white arrow marks the onset of mCherry expression. Scale bar, 50 μm. **c** Operetta measurements of activated hypnozoites with respect to EEF fluorescent surfaces (left panel) and mCherry intensity (right panel) at different days post sporozoite (spz) inoculation (green lines). For reference, the measurements of five individual primary liver stage schizonts as already are also depicted (red lines).

measured over time. Comparing this to measurements performed on several primary schizonts shows that the activated hypnozoites in untreated wells (Fig. 4c, green lines) resume development after the primary schizonts have burst (Fig. 4c, red lines and Supplementary movie 4). Furthermore, this reveals that early hypnozoite activation, as defined by parasites initiating mCherry expression and growth, is already evident between days 12–16 after sporozoite invasion. The speed of schizont maturation of activated hypnozoites appears to be similar to that of primary schizonts (Fig. 4c).

## Discussion

In our hands, *P. cynomolgi* liver stage cultures could be routinely maintained at good quality for 3 weeks. Thereafter, cultures often started to deteriorate, and this is why we focused on reactivation events during the first 3 weeks of culture. The few hypnozoite reactivation events measured in vitro (approximately one spontaneous event per 400 hypnozoites) suggest that only a minority of hypnozoites is prone to become activated at a certain time within the first 3 weeks of culture. In vivo, a similar phenomenon was observed when monkeys were inoculated with low numbers of *P. cynomolgi* sporozoites[8,28]. Infection with 500 sporozoites induced multiple relapses in all monkeys infected, whereas monkeys infected with only five sporozoites induced a primary infection in 3/5 monkeys, but relapses were not observed[8]. Monkeys receiving 10–20 sporozoites all developed a patent parasitemia, and 3/8 monkeys relapsed over an observation period of 1 year[28]. Interestingly, only one monkey showed an early relapse, at day 30 post infection. The second monkey relapsed at day 168 and the third monkey relapsed at days 58 and 75 post inoculation[28]. This suggests that some sporozoites may be prone to activate early, whereas others may be prone to activate late. This may have a biological reason, as the parasite's main goal is to be transmitted to the next host. Activation of most or all hypnozoites at once may not

be the best strategy to be transmitted in areas of low, unstable transmission, because there would be only a single opportunity to be transmitted. For human malaria in tropical regions, the ability to repeatedly induce new blood stage infections may be necessary to compete with more virulent *Plasmodium falciparum*[29]. Instead of a predetermined duration of dormancy, reactivation of hypnozoites may occur stochastically. To study possible patterns in reactivation, culture conditions will have to be further optimized to obtain healthy hepatocyte cultures lasting for at least 2 months.

Despite the global importance of relapse infections, there is little knowledge on basic hypnozoite biology. We anticipate that the dual fluorescent parasite line we describe here will be a valuable tool to increase our knowledge on hypnozoite dormancy and activation mechanisms. This parasite not only allows real-time monitoring of parasite development, but also enables FACSsort purification of reactivating hypnozoites for molecular characterization, as an addition to previously described hypnozoite purification with another transgenic *P. cynomolgi* line[16]. Furthermore, the ability to now quantitate hypnozoite activation provides the opportunity to evaluate compounds that trigger hypnozoite activation. If we can identify compounds that provoke hypnozoites to resume development, we can then eliminate the resulting schizonts with already available liver stage schizonticides in so-called 'wake-and-kill' strategies. This opens up new avenues to eliminate the hypnozoite reservoir from the human population, a key road block for malaria elimination.

## Methods
### Key resources table

| Reagent or resource | Source | Identifier |
|---|---|---|
| **Antibodies** | | |
| Rabbit anti-*P. cynomolgi* HSP70.1 polyclonal antibody | Ref. [17] | N/A |
| Alexa Fluor 647-conjugated Goat anti-Rabbit IgG (H + L) | Thermo Fisher Scientific | Cat. #A21245 |
| **Bacterial and virus strains** | | |
| Subcloning efficiency™ DH5α competent cells | Thermo Fisher Scientific | Cat. #18265-017 |
| **Biological samples** | | |
| Human serum A+ (pooled, heat inactivated) | Sanquin blood bank | N/A |
| FBS | Greiner Bio-One | Cat. #758093 |
| Primary rhesus hepatocytes | BPRC | N/A |
| **Chemicals, peptides, and recombinant proteins** | | |
| William's E | Thermo Fisher Scientific | Cat. #32551-087 |
| 100 × Pen/strep | Thermo Fisher Scientific | Cat. #15140-122 |
| 100 mM sodium pyruvate | Thermo Fisher Scientific | Cat. #11360-036 |
| 100 × Insulin/transferin/selenium supplement | Thermo Fisher Scientific | Cat. #41400-045 |
| 100 × MEM-NEAA | Thermo Fisher Scientific | Cat. #11140-035 |
| 0.1 M Hydrocortisone | Sigma | Cat. #H0888 |
| 2-Mercaptoethanol | Thermo Fisher Scientific | Cat. #31350-010 |
| DMSO hybrimax | Sigma | Cat. #P1860 |
| Pyrimethamine | Sigma | Cat. #P7771 |
| Matrigel | Corning | Cat. #354234 |
| Nycodenz | Axis Shield | Cat. #1002424 |
| Leibovitz L15 medium | Thermo Fisher Scientific | Cat. #11415-056 |
| Atovaquone | Sigma | Cat. #A7986 |
| KDU691 | Novartis | N/A |
| 4% Paraformaldehyde in PBS | Affymetrix | Cat. #J19943 |
| 4′,6-diamidino-2-phenylindole, dilactate (DAPI, dilactate) | Thermo Fisher Scientific | Cat. #D3571 |
| Hoechst33342 | Thermo Fisher Scientific | Cat. #H3570 |
| **Critical commercial assays** | | |
| QIAfilter Plasmid Maxi Kit | Qiagen | Cat. #12263 |
| Human T Cell Nucleofector Kit | Lonza | Cat. #VPA-1002 |

(continued)

| Reagent or resource | Source | Identifier |
|---|---|---|
| **Deposited data** | | |
| **Experimental models: cell lines** | | |
| Parasite strain: *Plasmodium cynomolgi* M | Ref. [30] | N/A |
| **Experimental models: organisms/strains** | | |
| Monkey: *Macaca mulatta* | In house breeding colony | N/A |
| Mosquito: *Anopheles stephensi* Sind–Kasur strain Nijmegen | RUNMC Nijmegen, NL | N/A |
| **Oligonucleotides** | | |
| **Recombinant DNA** | | |
| pCR-BluntII-PcyCEN | Ref. [16] | N/A |
| Building block from plasmid # 1 | This paper | N/A |
| Building block from plasmid # 2 | This paper | N/A |
| Building block from plasmid # 3 | This paper | N/A |
| pCyCEN_Lisp2mCherry_hsp70_GFP | This paper | Addgene #137169 |
| **Software and algorithms** | | |
| Las X | Leica | https://www.leica-microsystems.com/products/microscope-software/p/leica-las-x-ls/ |
| GraphPad Prism 7 | GraphPad software | www.graphpad.com |
| Columbus 2.8.2 | PerkinElmer | www.perkinelmer.com |
| **Other** | | |
| CellCarrier-96 Ultra Microplates, collagen coated | PerkinElmer | Cat. #6055700 |

**Experimental model and subject details**. *Nonhuman primates*: Nonhuman primates were used because no other models (in vitro or in vivo) were suitable for the aims of this project. The research protocol was approved by the central committee for animal experiments (CCD license number AVD5020020172664) and the subprotocol was approved by the local independent ethical committee constituted conform Dutch law (BPRC Dier Experimenten Commissie, DEC; agreement number #708 and #007 C) prior to the start of the experiments. All experiments were performed according to Dutch and European laws. The Council of the Association for Assessment and Accreditation of Laboratory Animal Care (AAALAC International) has awarded BPRC full accreditation. Thus, BPRC is fully compliant with the international demands on animal studies and welfare as set forth by the European Council Directive 2010/63/EU, and Convention ETS 123, including the revised Appendix A as well as the 'Standard for humane care and use of Laboratory Animals by Foreign institutions' identification number A5539-01, provided by the Department of Health and Human Services of the United States of America's National Institutes of Health (NIH) and Dutch implementing legislation. At least once a year, the health of the animals is checked and prior to experimentation an additional health check is carried out, including a physical examination and performing hematological, clinical-chemistry, serology, and bacteriological and parasitological analyses. Only healthy animals were included in the experiments. The rhesus monkeys used in this study (*Macaca mulatta*, either gender, age 4–16 years, Indian origin) were captive-bred and socially housed. Animal housing was according to international guidelines for nonhuman primate care and use. Besides their standard feeding regime and drinking water ad libitum via an automatic watering system, the animals followed an environmental enrichment program in which, next to permanent and rotating nonfood enrichment, an item of food enrichment was offered to the macaques daily. All animals were monitored daily for health and discomfort. Monkeys were trained to voluntarily present for thigh pricks, and were rewarded afterward. All intravenous injections and large blood collections were performed under ketamine sedation, and all efforts were made to minimize suffering. Liver cells were derived from inhouse frozen batches of hepatocytes or from freshly collected liver lobes from monkeys that were euthanized in the course of unrelated studies (ethically approved by the BPRC DEC) or euthanized for medical reasons, as assessed by a veterinarian. Therefore, none of the animals from which liver lobes were derived were specifically used for this work, according to the 3R rule thereby reducing the numbers of animals used. Euthanasia was performed under ketamine sedation (10 mg/kg) and was induced by intracardiac injection of euthasol 20%, containing pentobarbital.

*Mosquitoes*: Mosquito infections were performed as described previously[17]. *Anopheles stephensi* mosquitoes Sind–Kasur strain Nijmegen[31] were obtained from Nijmegen UMC, The Netherlands. At peak parasitemia, around 12 days after infection with blood stage *P. cynomolgi* M strain parasites, monkeys were bled for ex vivo mosquito feeding. Two to five-day-old female mosquitoes were fed on the infected blood, using a water-jacketed glass feeder system kept at 37 °C, female mosquitoes were fed on blood obtained from a monkey that had been infected with *P. cynomolgi* M strain blood stage parasites. Mosquitoes were housed for ~3 weeks in climate chambers at 25 °C and 80% humidity, and fed daily via

cotton soaked in 5% D-glucose solution. One week after infection, oocysts were counted and mosquitoes were given an additional uninfected blood meal to promote sporozoite invasion of the salivary glands.

*Malaria parasites*: *P. cynomolgi* M strain stocks were originally provided by Dr. Bill Collins from the Center for Disease Control, Atlanta, USA[32]. Stocks were maintained as frozen stabilates in cryoprotectant (28% glycerol, 3% sorbitol, and 0.65% NaCl in water) in liquid nitrogen. Parasites were thawed in a waterbath at 37 °C followed by sequential wash steps with equal volumes of 3.5% NaCl[33]. Parasites resuspended in phosphate-buffered saline (PBS) for i.v. injection ($10^6$ parasites in 1 ml PBS) into a monkey.

*Primary rhesus hepatocyte cultures*: Primary hepatocyte cultures were initiated either from freshly isolated *M. mulatta* hepatocytes through collagenase perfusion[34], or from in-house cryopreserved stocks. Hepatocytes were seeded into collagen coated 96-well CellCarrier Ultra plates (PerkinElmer) at a density of ~$65 \times 10^3$ cells/well in William's B medium: William's E with glutamax containing 10% human serum (A+), 1% MEM nonessential amino acids, 2% penicillin/ streptomycin, 1% insulin/transferrin/selenium, 1% sodium pyruvate, 50 μM β-mercapto-ethanol, and 0.05 μM hydrocortisone. Following cell attachment, medium was replaced by William's B containing 2% dimethylsulfoxide (DMSO) to prevent hepatocyte dedifferentiation. Prior to adding sporozoites, cells were washed twice in William's B medium. Liver stage cultures were maintained in William's B medium in a humidified incubator at 37 °C and at 5% $CO_2$.

**Method details**. *Plasmid DNA cloning*: Building blocks for the final construct were synthesized and cloned into PUC18, PUC19, or PUC57 mini plasmids by Genscript (USA). See the Supplementary Data file 1 for an outline of the cloning strategy and for sequences of the building blocks. Briefly, fragment Nluc-2A-mCherry was excised (*Eco*RV/*Bgl*II) from plasmid #1 and ligated to the *Eco*RV/*Bgl*II-digested plasmid #2 containing flanking regions of *P. cynomolgi lisp2* (PcyM_0307500) to generate plasmid (a). The *lisp*2-mCherry_2A_Nluc expression cassette was cloned into the *Bam*HI/ *Kpn*I sites of building block plasmid #3 to generate plasmid (b). This plasmid was linearized with *Not*I and ligated to *Not*I-linearized plasmid pCR-BluntII-TOPO containing a *P. cynomolgi* centromere[16] (Genbank accession number JQ809338) to create construct pCyCEN_Lisp2mCherry_hsp70_GFP (Addgene, ID 137169). All transformations were performed using subcloning efficiency DH5alpha cells (Invitrogen) according to the manufacturer's instructions. For transfection, maxiprep DNA was isolated (Qiagen).

*Parasite transfection*: Parasite transfection was essentially as described before[16]. A donor monkey was infected (i.v.) with $10^6$ *P. cynomolgi* M wild-type blood stage parasites from a cryopreserved stock. Parasitemia was monitored by reading Giemsa stained thin blood films prepared from thigh pricks. At a peak parasitemia of ±1% trophozoites, heparin blood was obtained and the monkey was cured from malaria by chloroquine treatment (i.m. 7.5 mg/kg) on three consecutive days. For animal welfare reasons, we have previously developed a Nycodenz protocol to enrich for *P. cynomolgi* blood stage parasites needed for transfection[16]. Using this method, parasite preparations of sufficient purity for transfection can be prepared, avoiding the need for splenectomy of the donor monkey in order to obtain high yields of parasites. To this end, the infected blood was layered on top of a 55% Nycodenz/PBS cushion and centrifuged (25 min, 300×*g*, RT, low brake). The interface containing ±30% trophozoites was isolated and washed in RPMI. Parasites were cultured at 2% hematocrit overnight in complete medium (RPMI-1640 containing 20% heat inactivated human A+ serum and 15 μg/ml gentamicin) at 37 °C and gas conditions of 5% $CO_2$, 5% $O_2$, and 90% $N_2$. The next day, medium was refreshed and the culture was maintained until fully matured schizonts were observed by Giemsa stained thin films of the culture. To increase levels of successful transfection, two methods were used. Following a wash step in RPMI-1640, ±$3 \times 10^8$ parasites were gently resuspended in Cytomix (120 mM KCl, 0.15 mM CaCl$_2$, 2 mM EGTA, 5 mM MgCl$_2$, 10 mM K$_2$HPO$_4$, 10 mM KH$_2$PO$_4$, 25 mM Hepes, pH 7.6) containing 45 μg plasmid DNA. The mixture was transferred to a 4 mm electroporation cuvette (Bio-Rad), electroporated at 25 μF, 2500 V and 200 Ω, and put on ice until injection. In addition, $2 \times 10^7$ parasites were gently resuspended in 100 μl Human T-Cell Nucleofector solution (Lonza), mixed with 10 μg plasmid DNA and electroporated using a Nucleofector device (Lonza, program U-033). The mixtures from the two transfection methods were combined in 0.5 ml PBS and i.v. injected into a recipient monkey. At day 4 post transfection, blood stage parasites were observed in the recipient monkey. From the next day onward (four times), the recipient monkey received every other day pyrimethamine (1 mg/kg) hidden in a piece of fruit to select for transfectants. First parasites were observed at day 11 post transfection. At peak parasitemia, blood was taken for mosquito feeding, stocks, and analyses. Subsequently, the monkey was cured from malaria by chloroquine treatment (i.m. 7.5 mg/kg) on three consecutive days. For mosquito feeding, recipient monkeys were infected with $1 \times 10^6$ transgenic *P. cynomolgi* M strain parasites from a cryopreserved stock. To eliminate possible wild-type contaminant parasites, the monkeys received three dosages of pyrimethamine (1 mg/kg, orally in a piece of fruit every other day), starting one day post infection. Around peak parasitemia, on two consecutive days, generally around days 11–13 post infection, 5–9 ml of heparin blood was taken to feed mosquitoes and monkeys were cured from *Plasmodium* infection by intramuscular treatment with chloroquine (7.5 mg/kg) on three consecutive days. Experiments

were performed entirely independently from each other: parasites were isolated from different donor monkeys; transmission experiments and hepatocyte cultures were performed at different time points.

*Live imaging of P. cynomolgi infected blood stage parasites*: At the time of ex vivo mosquito feeding using blood from a monkey infected with the transgenic *P. cynomolgi* line, a drop of blood was mixed 1:1 with Hoechst 33342 (resulting in a final concentration of 10 μg/ml) and viewed under a Leica DMI6000B inverted fluorescence microscope HC PL APO 63x/1.40–0.60 oil objective and pictures were acquired with a DFC365FX camera.

*P. cynomolgi sporozoite isolation and liver stage culture*: Two weeks post mosquito feeding on transgenic *P. cynomolgi* M strain-infected blood, salivary gland sporozoites were isolated and used for hepatocyte inoculation. Salivary glands were collected on ice in Leibovitz's medium containing 3% fetal calf serum and 2% Pen/Strep. Salivary glands were disrupted using a Potter-Elvehjem homogenizer and debris was removed by a slow spin in a microfuge (3 min, 60×*g*, RT), before counting the sporozoites in a Bürker-Türk counting chamber. Primary rhesus hepatocytes seeded in 96-well CellCarrier Ultra plates (PerkinElmer) two days earlier were inoculated with $5 \times 10^4$ sporozoites per well in William's B medium. Plates were spun for 5 min at 233×*g* (RT, low brake) before transfer back into the humidified incubator (37 °C and at 5% $CO_2$) for 2–3 h to allow for sporozoite invasion. Plates were washed with William's B medium and for prolonged cultivation, 80 μl of Matrigel (Corning) was added on top of the infected hepatocyte monolayer. After 30 min solidification at 37 °C, William's B medium was added and incubation was continued with regular medium refreshments until fixation with 4% paraformaldehyde (PFA) for 30 min at RT. For drug treatment, *P. cynomolgi* liver stage cultures were daily treated with 0.5 μM KDU691 (Novartis) in William's B medium from days 5 to 8.

*Live imaging of P. cynomolgi infected hepatocytes*: Images of transgenic *P. cynomolgi* parasites were acquired using a Leica DMI6000B inverted fluorescence microscope equipped with a DFC365FX camera and a HC PL APO 63x/1.40–0.60 oil objective or a HC PL APO 40x/1.30 oil objective. For monitoring liver stage development over time, daily or every other day imaging was performed using an Operetta High Content Imaging system (PerkinElmer) and a 20x long WD objective. Acquisition times were kept constant. To determine whether the frequent imaging had any effect on parasite growth, images of parasites that had been imaged five times up to day 10 post infection were compared to parasites that were only imaged once at day 10. No differences in parasite sizes or numbers were observed indicating that repeated imaging apparently was not detrimental to the parasites.

To determine the number of reactivating hypnozoites versus nonreactivated hypnozoites, the number of day-15 hypnozoites was determined in five independent infections by counting five wells, one well of each infection. This resulted in an average of 56 day-15 hypnozoites per well, equaling 1176 hypnozoites in 21 wells.

*Analysis of parasite size and fluorescence intensity*: Parasite size and fluorescence intensities were determined using a customized Columbus script. For detection of live *P. cynomolgi* liver stage forms the following criteria were used: threshold for GFP expression was 0.50 and GFP-positive populations were counted when the mean GFP intensity was >600. When confirmed manually that parasites were identified by the script, the fluorescence and size characteristics of the parasites were recorded. The characteristics of the same parasites were determined at various time points during liver stage culture.

*Immunofluorescence analysis*: Liver stage cultures were fixed in 4% PFA for 30 min at RT and overnight incubated at 4 °C with rabbit anti-*P. cynomolgi* HSP70 primary antibody diluted 1:10,000 in antibody dilution buffer (0.3% Triton-X100, 1% bovine serum albumin in PBS). Samples were washed with PBS and incubated for 2 h at RT with Alexa 647-conjugated goat-anti-rabbit IgG (Thermo Fisher Scientific, 1:1000) and 2 μM DAPI (Thermo Fisher Scientific) in antibody dilution buffer. Samples were washed again in PBS and images were captured on a Leica DMI6000B inverted fluorescence microscope equipped with a DFC365FX camera.

**Statistics and reproducibility**. *Sample sizes and statistical analysis*: Data were analyzed using Prism 7.0 (GraphPad Software) or Columbus 2.8.2 (PerkinElmer). Results are presented as *n* values (the number of wells analyzed) ± s.e.m. as described in the figure legends.

**Reporting summary**. Further information on research design is available in the Nature Research Reporting Summary linked to this article.

## Data availability

The data, resources, and reagents that support the findings of this study are available from the corresponding author upon reasonable request. The transfection construct pCyCEN_Lisp2mCherry_hsp70_GFP can be requested through Addgene, ID 137169.

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

## Acknowledgements

We are grateful to the members of the Animal Science Department for excellent animal care and veterinary assistance. We thank Anke Harupa for expert help in monitoring the exflagellation process, Francisca van Hassel for preparing graphical representations and the mosquito breeding facilities in Nijmegen for provision of *A. stephensi* mosquitoes. This work was funded by the Bill & Melinda Gates Foundation (OPP1141292).

## Author contributions

Conceptualization, A.V. and C.K.; methodology, A.V., C.K. and A.M.Z.; investigation, A.V., A.M.Z., I.N., N.W., E.K., O.K. and L.V.; resources, D.K.G., L.D. and T.T.D.; writing—original draft, A.V. and C.K.; writing—review and editing, A.V., A.M.Z., D.K.G., L.D., T.T.D. and C.K.

## Competing interests

Devendra Kumar Gupta and Thierry Tidiane Diagana are employed by and/or are shareholders of Novartis Pharma AG. The other authors declare that no competing interests exist.
