## [Peer Review File · Communications Biology]

Reviewers' comments:

Reviewer #1 (Remarks to the Author):

Voorberg-van der Wel and colleagues describe the generation of a dual reporter line for analysis of *Plasmodium cynomolgi* liver stage parasites. *P. cynomolgi* is biologically and genetically similar to *P. vivax* and is a good model system to study non-replicating liver stage parasite forms hypnozoites. Putting mCherry under a liver stage specific promoter (LISP2) is an excellent idea to visualize developing liver stage parasites. The authors find that hypnozoites stages can be activated in in vitro culture over time, which is an exciting and novel finding.

- 1) Fig 2. And Supplementary fig 2. The authors mention that onset of mCherry expression under LISP2 promoter corresponded with nuclear division. The supplementary figure 2 depicting this is not included in the manuscript. Authors can include this figure in the manuscript. Panel for DAPI stained images for figure 2a should be incorporated.
- 2) Fig 4. The authors use a PI4K inhibitor to specifically eliminate schizonts from the culture. Authors may include some of the images in the manuscript before and after treatment of parasites with the inhibitor. Have authors collected and analyzed if merozoites coming out of activated hypnozoites are infectious?
- 3) Line 289. Was D-Glucose solution used for feeding mosquitoes supplemented with Para-aminobenzoic acid (PABA)?
- 4) Line 337. Authors may cite appropriate reference.
- 5) Line 358. Parasite stocks and analysis. Parasite genotyping details should be included in the manuscript. Were the dual reporter parasites cloned?
- 6) Statistical methods used for analysis of data for all the figures should be included in the figure legends.

Reviewer #2 (Remarks to the Author):

Brief summary of the manuscript

The reactivation of *Plasmodium vivax* hypnozoites in the liver is purportedly the cause of the majority of vivax malaria blood stage infections. The authors of this manuscript utilize the *P. cynomolgi* simian malaria model system and imaging technologies to show the reactivation of these dormant liver stage parasites. This manuscript describes the development and testing of a *P. cynomolgi* transgenic parasite line containing a centromere plasmid that expresses *gfp* and the human *dhfr* selectable marker under the control of the constitutively expressed *P. cynomolgi* *hsp70* promoter, plus mCherry upon reactivation of hypnozoites. mCherry expression is under the control of the promoter and 3'UTR sequences of the *P. cynomolgi* schizont-specific marker for early liver stage growth, otherwise known as liver-specific protein 2 (*lisp2*).

Transgenic and wild type parasites were assessed side by side, with no fitness loss noted, and the full cycle was documented showing GFP expressed in invasive merozoites, blood stages, gametocytes, mosquito oocysts, salivary gland sporozoites, and then liver stage forms that developed into schizonts and ultimately released invasive merozoites from merozoites, which were shown to be able to infect red blood cells. Several movie clips were included to show GFP expression with merozoite invasion, exflagellation, schizont rupture, and merozoite release of new fluorescent merozoite progeny. The centromere plasmid was maintained by a majority (73%) of the liver stage parasites, which were not under pyrimethamine selection, and this is comparable to previous published work with centromere plasmids.

Live fluorescence microscopy was performed to show the stages of development and expression and stage specificity of the GFP and mCherry markers. Specific individual parasites were observed and monitored daily or every other day with regards to their development and GFP or mCherry expression using an Operetta high content imager.

The progression of the in vitro liver stage infections is shown and described in a clear manner, and the data indicate the expected time frame for initial stage-specific development, and then convincingly show at a later point in time, the reactivation of forms that remain small. Some parasites that initially remain as hypnozoites with GFP expression, later display new developmental activity with the expression of mCherry and ultimately schizont formation with merozoites released. Established PI4K inhibitor drug treatment was applied to ensure elimination of primary schizonts, and the strict monitoring of hypnozoites and their reactivation.

The authors detail technical matters that could potentially be improved upon, or understood better with further investigation; e.g., the reduction of parasites noted from day 10 to 15 (Supp Table 1). In addition, in the in vitro system used by these investigators, over a 3-week period, only a small number of confirmed hypnozoites were shown to reactivate and form secondary schizonts with the release of merozoites (only 3 out of 1175 day-15 hypnozoites monitored; or approximately 1 spontaneous event per 400 hypnozoites). More activation would be advantageous for follow-up investigation of these parasites, both hypnozoites and the reactivated forms. One cannot say for sure how this compares with in vivo infections, and whether this is a limitation of their specific in vitro culture system; i.e., can the number of reactivation events within 22 days be increased in vitro? As the authors note, hepatocyte culture systems that thrive for longer periods of time will be required for more in depth analyses, and which also support the population of hypnozoite parasites through until their eventual reactivation, whether a few days, weeks or months later.

Overall impression of the work

This manuscript is of global importance, pertinent to research aiming to eradicate malaria, and specifically relapsing malaria parasites. Overall, the work is well done, and represents an important step forward in this critical and challenging area of research. The design and presentation of the results are clear, and consistent with the study goals and reported findings. The low yield of reactivating hypnozoites achieved in this study is important to report and take into consideration for future studies, whether with repeat or modified culture efforts with this or other liver culture systems, when ideally extending the infected liver cell culture time frame beyond 22 days - if possible - to observe more reactivation events, for experiments aimed to study the biology of these forms, or modeling efforts to relate the results to the in vivo situation and epidemiology of the disease.

Specific comments, with recommendations for addressing each comment

1. I suggest re-wording the phrase "repeat episodes of disease relapses", as is noted in the beginning of the abstract, as hypnozoite reactivation and relapses do not necessarily in all cases lead to clinical disease with symptoms. It is important to make this distinction.
2. Similarly, to be correct and consistent with reference #8, the first sentence should refer to *P. vivax* infections, not cases.
3. Members of this team previously reported longer term cultures up to 40 days (Dembele et al. 2014). A comment on the experimental design and rationale here for running the experiments for only 22 days would be useful, particularly since longer cultures are a recognized requirement for furthering studies of relapsing parasites in vitro, and the number of reactivating hypnozoites is low in the 22-day time frame. What happens at day 23? And beyond?
4. With more time in culture, will the centromere plasmid be maintained to be able to continue to monitor the presence and activation of these parasites using the transgenic fluorescence markers? Integrated transgenes may become essential.

5. Some discussion has been provided on relating the in vitro reactivation data to a few published in vivo *P. cynomolgi* relapse studies, across which the sporozoite inoculum number has been quite variable, ranging from low numbers closer to human infections to higher experimental numbers that better ensure relapses within the experimental time frames. There are a few other recent in vivo studies that would seem worth mentioning as well, including Deye et al. 2012 and Joyner et al. 2016/2019, particularly in light of the limited body of research in this area, and the interest in relating the in vitro findings to in vivo infections. Together, these studies show that higher sporozoite inoculums result in earlier initial relapses.

6. Line 33 notes that "people have questioned the theory of relapse". Perhaps say "over time, a number of researchers have questioned..."? Or, "The theory of relapse has been questioned (refs)". Regardless, Dembele et al 2014 reported the reactivation of apparent hypnozoites, in support of the relapse theory, with hypnozoite reactivation. Still and all, this paper is the first proof, I believe, showing the reactivation of specific latent infected hepatocytes, with the benefit of real time monitoring over time.

7. Line 336-337, numbered reference needs to be added.

8. Line 337, preps should be changed to preparations.

9. Line 551, parasites is written twice.

Response to the referees' comments (manuscript COMMSBIO-19-1299A):

We would like to thank the editor and reviewers for their thorough review of our manuscript. We are grateful for the positive responses of the referees. Below we address the points raised.

Reviewer #1:

1) Fig 2. And Supplementary fig 2. The authors mention that onset of mCherry expression under LISP2 promoter corresponded with nuclear division. The supplementary figure 2 depicting this is not included in the manuscript. Authors can include this figure in the manuscript. Panel for DAPI stained images for figure 2a should be incorporated.

The reviewer requests that a panel for DAPI should be incorporated. Fig 2a and 2b derive from different experiments. In Fig 2a we show high magnification pictures of the GFP/mCherry expression in the parasites. During this experiment we attempted to follow parasites over time with a regular fluorescent microscope and therefore we did not include DAPI because we did not want to disturb the parasite growth.

The data from Fig 2b come from a different experiment in which we quantitated the GFP/mCherry signals in relation to the parasite nuclei. We agree with the reviewer that it is important to illustrate this in a picture that shows the DAPI signals in combination with GFP/mCherry. That is why we included Supplementary figure 2, but apparently, something has gone wrong during the submission process, resulting in the omission of Supplementary figure 2 and this has escaped our attention. We have now corrected this and show in Supplementary figure 2 mCherry expression only in multinucleate schizonts.

2) Fig 4. The authors use a PI4K inhibitor to specifically eliminate schizonts from the culture. Authors may include some of the images in the manuscript before and after treatment of parasites with the inhibitor. Have authors collected and analyzed if merozoites coming out of activated hypnozoites are infectious?

We agree with the reviewer that one of the strengths of the fluorescent line is that it enables follow up of parasites under drug pressure. To illustrate this, we have now included some pictures in the Supplementary Information file (Supplementary figure 3). This shows parasites before and after drug treatment, demonstrating that schizonts are eliminated as a result of the drug treatment.

We have not collected merozoites coming out of activated hypnozoites to analyze whether they are infectious. This is technically demanding for a number of reasons. In primary cultures with high numbers of merozoites the efficiency of blood stage infection appeared low. This may have to do with the time of adding the RBC (merozoites may not be viable for prolonged times; the RBC age of the added RBC may not be optimal). As we have shown, the number of activated hypnozoites is very low, so the number of merozoites is very limited. Furthermore, optimal timing of adding RBC is likely important and given the time consuming (manual) search for activated hypnozoites this is very difficult to achieve.

3) Line 289. Was D-Glucose solution used for feeding mosquitoes supplemented with Para-aminobenzoic acid (PABA)?

The glucose solution for feeding was not supplemented with any substance. Indeed, it has been shown that PABA can enhance the sporogonic development of avian (Terzian LA et al, J Inf Dis 90:115-123) and rodent malarial (Peters W and Ramkaran AE, Ann Trop Med Parasitol 74:275-282). However, for *P. falciparum* PABA supplementation did not enhance parasite development (Beier, MS

et al, J Med Ent 31(4):561-565) and this may point to a difference between primate malarias and avian/rodent malarias. We have not tested this for *P. cynomolgi* because we are satisfied with the sporozoite numbers we routinely obtain in the salivary glands (on average 50,000 spz or more per mosquito; see Supplementary fig1).

4) Line 337. Authors may cite appropriate reference.

We thank the reviewer for pointing out this omission. We have now corrected this.

5) Line 358. Parasite stocks and analysis. Parasite genotyping details should be included in the manuscript. Were the dual reporter parasites cloned?

The parasites were not cloned. This is complicated because this is an *in vivo* system using rhesus monkeys. Cloning would involve multiple animals which is difficult to ethically justify. The transfection efficiency of *P. cynomolgi* is estimated to be very low, so it is anticipated that only limited numbers of parasites will have taken up DNA and this may have resulted in a limited genetic variability. Given the clearly demonstrated phenotype, we did not genotype the parasites. Our main interest was to be able to follow the parasite development inside liver cells and the parasites we developed served this purpose well.

6) Statistical methods used for analysis of data for all the figures should be included in the figure legends.

This manuscript reflects an observational study, thus involving only very limited statistical analyses. Where appropriate we have mentioned sample size, and other statistical characteristics such as mean, median, SEM values. We are not aware of any missing statistical data, but we are open to hear from the reviewer what statistical information we would need to add (and where).

Reviewer #2:

Specific comments, with recommendations for addressing each comment

1. I suggest re-wording the phrase “repeat episodes of disease relapses”, as is noted in the beginning of the abstract, as hypnozoite reactivation and relapses do not necessarily in all cases lead to clinical disease with symptoms. It is important to make this distinction.

We thank the reviewer for pointing this out. This is a good point and we completely agree with this comment. We have amended the text. It now reads: *Plasmodium vivax malaria is characterized by repeated episodes of blood stage infection (relapses).*

2. Similarly, to be correct and consistent with reference #8, the first sentence should refer to *P. vivax* infections, not cases.

The reviewer is correct. We have now re-phrased this sentence: *The majority of *P. vivax* infections is likely due to relapses⁸.*

3. Members of this team previously reported longer term cultures up to 40 days (Dembele et al. 2014). A comment on the experimental design and rationale here for running the experiments for only 22 days would be useful, particularly since longer cultures are a recognized requirement for

furthering studies of relapsing parasites in vitro, and the number of reactivating hypnozoites is low in the 22-day time frame. What happens at day 23? And beyond?

We agree with the reviewer that prolonged culturing would provide more insights into the kinetics of hypnozoite activation. Indeed, a longer-term *P. cynomolgi* culture of up to 40 days has been reported by Demebele et al. (Nat Med, 2014). However, only a limited set of experiments on the culture for 40 days were reported and this was in the absence of Matrigel (Figure 1 of that paper). Later in the paper it was shown that Matrigel was needed to preserve hepatocyte quality. Only one culture at 25 days was shown with very limited numbers of parasites. It is therefore not clear whether parasites can routinely be maintained for longer times than 3 weeks. In our hands, Matrigel significantly improved the culture quality as assessed by looking at hepatocyte morphology. However, even in the presence of Matrigel the culture appeared to start deteriorating after more than 3 weeks, and that is why we did not look at later time points. We also see differences in the capacity for prolonged cultivation between batches of hepatocytes. We think more optimization of culture conditions is needed for routine maintenance of good quality long term cultures. This may require 3D systems, such as organoids. As suggested by the reviewer, we have now included a sentence (line 195) on the rationale of maintaining the cultures for only 3 weeks:

In our hands, P. cynomolgi liver stage cultures could be routinely maintained at good quality for 3 weeks. Thereafter, cultures often started to deteriorate, and this is why we focused on reactivation events during the first 3 weeks of culture.

4. With more time in culture, will the centromere plasmid be maintained to be able to continue to monitor the presence and activation of these parasites using the transgenic fluorescence markers? Integrated transgenes may become essential.

It is known that episomes are unevenly distributed during mitosis and that subsequent drug pressure is needed to select for parasites that still have the episome. The presence of the centromere has enabled a more stable presence of the plasmids, even after multiple rounds of division (Barale and Menard, (2010), CHM 7:181-183).

Even after multiple rounds of division without drug selection in the mosquito, 73% of the liver stage parasites still express the fluorescent reporters. Only when new parasites (merozoites) have emerged following hypnozoite activation and schizont maturation, it is anticipated that a minor part of the merozoites will not have the centromere construct due to the uneven segregation. Therefore, we are not afraid that the capacity to observe hypnozoite activation in prolonged cultures will be lost.

However, we absolutely agree with the reviewer that integrated transgenes will be a great improvement. It will mean that all parasites express the transgene (compared to only 73% as shown here) increasing the sensitivity of the read out.

5. Some discussion has been provided on relating the in vitro reactivation data to a few published in vivo P. cynomolgi relapse studies, across which the sporozoite inoculum number has been quite variable, ranging from low numbers closer to human infections to higher experimental numbers that better ensure relapses within the experimental time frames. There are a few other recent in vivo studies that would seem worth mentioning as well, including Deye et al. 2012 and Joyner et al. 2016/2019, particularly in light of the limited body of research in this area, and the interest in relating the in vitro findings to in vivo infections. Together, these studies show that higher sporozoite inoculums result in earlier initial relapses.

We agree that it would be worthwhile to incorporate some more references on the *in vivo* relapse patterns. We have now done this and made some changes to the text (lines 146):

In vivo, *P. cynomolgi* M is a fast relapsing parasite. While low dose (2000 sporozoites) inoculations followed by subcurative treatment of blood stages gave rise to somewhat delayed relapses (ref Joyner), high dose sporozoite inoculations (10^5 - 10^6) resulted in early relapses around day 18-33 post infection as assessed by thin film analysis²⁹ (and ref Deye).

6. Line 33 notes that “people have questioned the theory of relapse”. Perhaps say “over time, a number of researchers have questioned...”? Or, “The theory of relapse has been questioned (refs)”. Regardless, Dembele et al 2014 reported the reactivation of apparent hypnozoites, in support of the relapse theory, with hypnozoite reactivation. Still and all, this paper is the first proof, I believe, showing the reactivation of specific latent infected hepatocytes, with the benefit of real time monitoring over time.

We thank the reviewer commenting positively on the strength of our paper. We have followed the suggestion of the reviewer and changed the sentence into:

*After the identification of hypnozoites in livers infected with *P. cynomolgi*^{10,11} and *P. vivax*³, the hypnozoite theory of relapse has been questioned¹²⁻¹⁴.*

7. Line 336-337, numbered reference needs to be added.

Thank you for noting this omission. We have now added the correct reference.

8. Line 337, preps should be changed to preparations.

We have now corrected this.

9. Line 551, parasites is written twice.

We have corrected this.

We have made another change in the legends of Figure 1, because we wanted to point out that Figure 1a is only a schematic figure of the most important elements of the construct, but that all elements are described elsewhere in the paper:

Schematic of the most important building blocks of the construct used for transfection; for details of the complete construct, see methods and Supplementary figure 4.

REVIEWERS' COMMENTS:

Reviewer #1 (Remarks to the Author):

The authors have properly addressed my queries and have made the appropriate changes to their manuscript which I look forward seeing published in Communications Biology.

Reviewer #2 (Remarks to the Author):

The authors have satisfactorily responded to my comments and suggestions.

Response to the reviewers' comments

Reviewer #1 (Remarks to the Author):

The authors have properly addressed my queries and have made the appropriate changes to their manuscript which I look forward seeing published in Communications Biology.

Reviewer #2 (Remarks to the Author):

The authors have satisfactorily responded to my comments and suggestions.

We thank both reviewers for their review of the paper. We are happy that the reviewers are satisfied with our response to their comments.